# Exploring Co-Regulation-Related Factors in the Mothers of ADHD Children—Proof of Concept Study

**DOI:** 10.3390/children10081286

**Published:** 2023-07-26

**Authors:** Ruth Yaacoby-Vakrat, Margalit Pade, Tami Bar-Shalita

**Affiliations:** Department of Occupational Therapy, The Stanley Steyer School of Health Professions, Faculty of Medicine, Tel-Aviv University, Ramat-Aviv, Tel-Aviv 6997801, Israel; ryaacoby@gmail.com (R.Y.-V.); pade@tauex.tau.ac.il (M.P.)

**Keywords:** executive functions, sensory processing, sensory modulation, scaffolding, mothers, self-efficacy, ADHD

## Abstract

Background: Attention deficit hyperactivity disorder (ADHD) is a complex neurological condition interfering with family relationships and co-regulation capabilities. Therefore, exploring factors underpinning parental co-regulation ability is crucial for future fostering relationships in families of children with ADHD. Objective: This preliminary study aims to characterize and compare the executive-functions, anxiety, self-efficacy, and sensory modulation in mothers of children with and without ADHD. Method: Mothers of children with (study group) and without (control-comparison, group) ADHD completed online self-reports, measuring executive-functions; parental self-efficacy; anxiety; and sensory modulation. Results: The study group (N = 40) had lower self-efficacy compared to the control group (N = 27; *p* = 0.018), and the control group had lower sensory responsiveness (*p* = 0.025). Within both groups the Behavior Rating Inventory of Executive Function-Adult (BRIEF-A) Global Executive Function Composite score (GEC) and the Beck Anxiety Inventory (BAI) were moderately correlated. Further, within the study group correlations were found between the BRIEF-A-GEC and the Sensory Responsiveness Questionnaire (SRQ)-Aversive scores (r = 0.37, *p* ≤ 0.01), and between the BRIEF-A Behavioral-Rating-Index and the parental self-efficacy scores (r = 0.31, *p* ≤ 0.05). Within the control group, negative correlations were found between the BRIEF-A-GEC and SRQ-Hedonic scores (r= −0.44, *p* < 0.05). Conclusion: Self-efficacy, executive-functions, high sensory responsiveness and anxiety traits are interwoven and may impact parental co-regulation ability.

## 1. Background

Attention deficit hyperactivity disorder (ADHD) is one of the most common neurodevelopmental disorders diagnosed in childhood [1], being characterized by persistent, disrupted patterns of hyperactivity, inattention, and/or impulsivity that interfere with development and daily functioning [2,3]. ADHD in childhood affects five to seven percent of all school-aged children [3,4,5], and comprises 33–50% of child referrals to mental health facilities [6]. Indeed, studies have documented a high rate of co-occurring psychiatric and learning disorders among these children [5,7]. ADHD is linked to poor response inhibition, thus interfering with children’s ability to develop higher executive functions (EFs) [8,9,10]. Compromised EFs can lead to reduced self-regulation [9], which negatively impacts the child’s functioning in school activities, self-care, play, leisure, and social participation [11].

Not surprisingly, parenting a child with ADHD is considered highly stressful and demanding [12,13]. Moreover, mothers have reported not only being less appreciated by family and teachers but also self-blamed. Feeling guilt negatively impacts mothering effectiveness [14], and mothers of children with ADHD receive less support in their daily routines [15]. Hence, mothering a child with ADHD consistently impacts child development [16], and often has adverse effects on mothers’ mental and emotional well-being [17]. Moreover, ADHD is familial by nature [7], with multiple etiologies including neurological, environmental, and genetic factors which contribute to its pathogenesis and thus to its heterogeneous phenotype [18]. Indeed, 25–30% of children with ADHD may have at least one parent diagnosed with ADHD [19,20]. Thus, the parent–child relationship is at risk becoming fundamentally impacted by ADHD [21]. Given that the involvement of a primary caregiver is crucial for successful treatment aimed at reducing ADHD symptoms and eliciting regulatory and functional capabilities [22,23,24,25,26,27], the support and strategies required by mothers of children with ADHD differ from those needed by mothers of children without ADHD [17,28,29].

Parental capabilities for regulating the behaviors of children with ADHD largely derive from their EFs [30,31]. EFs are considered higher cognitive abilities, enabling attentional and purposeful behavior in order to achieve functional goals [32]. Compromised EFs affect regulatory processes necessary for selecting, initiating, implementing, and overseeing thoughts, emotions, and behaviors [33]. Specifically, matching parental EFs to the demands placed on the child has been considered necessary to elicit motivation for goal achievement and autonomy mastering among children with ADHD [34,35]. When these are compromised, parents demonstrate a tendency to struggle with co-regulation and with scaffolding their children’s ADHD symptoms [30,34,36,37]. Scaffolding is defined as a process by which parents/ tutors help plan and organize children’s activities so that they can execute a task beyond their current level of ability. Scaffolding has been proposed as a mechanism accounting for the manner in which certain types of parent–child relationships facilitate children’s emotional development and self-regulatory capabilities [16,38,39], and is considered one of the foundations for the development of EFs in children [16]. Therefore, sufficient parental EFs are paramount, being required to facilitate the child’s ability to self-regulate his behavior, which in turn may elicit a better parenting performance [40,41]. However, compromised parental EFs have been linked to elevated anxiety, impacting the parent–child relationship [30,34,42,43], exacerbating the severity of the child’s ADHD symptoms, and leading to insufficient parental skills and practices [44]. Indeed, anxiety has been found to be one of the primary mental health states that can co-occur in parents and children with ADHD, with genetic links being observed between the two health conditions [5,45]. Hence, parents of children with ADHD experience more challenges in fostering a positive relationship with their child, and therefore are more preoccupied with and engaged in their children’s daily tasks [46].

Parental self-efficacy, a parent’s belief in his/her ability to successfully perform the parenting role [47], has been found to be a significant factor in successfully handling ADHD symptoms [24,48,49]. Further, it is crucial for developing regulatory abilities and enhancing child motivation [24,48]. Importantly, maternal stress impacts the perception of maternal roles and elicits increased feelings of low competence, negatively affecting the mother’s ability to support her child and thus disrupting the parent–child relationship [50]. Indeed, parents of children with ADHD report feeling incompetent and having lower parental self-efficacy in comparison with parents of children without ADHD [51]. Therefore, it is suggested that an intervention for children with ADHD should improve parental self-efficacy as a core skill, which will foster parents’ ability to manage their child’s ADHD symptoms [52]. In addition, parental self-efficacy is one of the main targeted goals crucial for refining family dynamics and enhancing regulatory capabilities [25,46]. The importance of self-efficacy in ADHD treatment is bidirectional, benefiting both the parents and the children with ADHD, as both demonstrate challenges in their daily activities and functioning [49,52,53].

Co-regulation challenges, particularly in parents of children with ADHD, are also impacted by sensory modulation dysfunction (SMD). Modulating the sensory input, the ability to respond to sensory stimuli adaptively and sufficiently, is a critical neurobehavioral process allowing the environment and surroundings to be comprehended, which is fundamental for the enablement of sufficient responses to encountered sensory stimuli [54,55]. SMD, a condition expressed by over- or under-responsiveness to daily sensations [55], is linked to high stress levels [56,57] and severely interferes with daily routines and participation as well as quality of life [49,52,53,58,59,60,61]. Furthermore, since 54–65% of children with ADHD demonstrate co-occurring SMD [19,62,63,64], their difficulties are amplified in sensory-intensified environments, such as schools [24,50,65]. Of note, like children with ADHD, children with SMD are reported to have mothers with SMD symptoms [66,67]. This suggests the parental sensory profile and the home environment are important contributors to the behavioral expressions of both mothers and children. However, to the best of our knowledge, the link between SMD and ADHD has not yet been explored in relation to parental EF, anxiety, self-efficacy, and co-regulation abilities.

Taken together, this preliminary study aimed to characterize and compare EF, self-efficacy, anxiety, and sensory modulation in mothers of children with and without ADHD. Deepening the understanding of parental characteristics and predispositions may have far-reaching implications for fostering healthy relationships, specifically co-regulation within families with children diagnosed with ADHD.

## 2. Methods

This preliminary study was a cross-sectional study carried out via an online survey that was distributed through social media networks. This study was approved by the Institutional Ethics Committee (#0002738-1), and all the anonymous participants consented to participation online before completing the survey.

### 2.1. Participants

Mothers of children with and without ADHD were invited to participate in the study, since mothers are considered the primary care provider [29].

Inclusion criteria stipulated mothers of children ages 6–12 with or without ADHD and with no other diagnosis.

### 2.2. Instruments

The Behavior Rating Inventory of Executive Function—Adult Version (BRIEF-A) [33] is a standardized self-report questionnaire comprising 75 items aiming to assess EF abilities in everyday environments, and thus is used in assessing functional cognition in individuals aged 18-90 years based on the past 30 days. The questionnaire contains nine scales, which together form two higher-order indexes corresponding to the Behavioral-Regulation Index (BRI), including (1) inhibition, (2) shifting, (3) emotional control, and (4) self-monitoring scales, to evaluate an individual’s ability to self-regulate behaviors and emotions, and the Metacognitive Index (MI), including the scales (5) initiation, (6) working memory, (7) plan/organize, (8) task monitor, and (9) organization of materials, to measure an individual’s ability to problem solve using consistent and systematic methods. Utilizing a three-point response scale, “never”, “sometimes”, or “often” (raw scores: 0–150), the BRIEF-A raw score is converted to a T-score, where 65 is the cut-off and higher scores indicate EF difficulties on each scale, index (BRI and MI), and the overarching summary score, the Global Executive Composite (GEC). Thus, lower scores indicate better functioning. The BRIEF-A has been validated in a variety of populations, such as in ADHD patients compared to non-clinical controls. The BRIEF-A has demonstrated excellent internal consistency (a Cronbach’s α of 0.93–0.94 for the three major indices) and one-month test–retest reliabilities (r = 0.93–0.94, three major indices) [33]. Convergent and divergent validity have been reported as well [68].

The Parental Self-Efficacy Questionnaire [69] is a standardized self-report questionnaire aiming to evaluate parents’ beliefs regarding their parenthood self-efficacy. The questionnaire describes different aspects of parental self-efficacy, including parent satisfaction with positive parental behaviors towards the child, feelings regarding their ability to handle parental challenges, difficulties implementing discipline, and being consistent in supporting their child. It also emphasizes additional feelings experienced towards the child. This questionnaire consists of 15 items, phrased as questions (e.g., *to what extent are you satisfied with your child-rearing skills*?) or statements (e.g., *I frequently reproof my child without a sensible reason*). Using a 6-point response scale (1—not at all to 6—very much), the total score is calculated as the sum of all item scores ranging from 5 to 90, where higher scores represent a greater level of parental self-efficacy. Factor analysis yielded four factors that accounted for 18.5% of the variance in the ability of a mother to set limits for her child. Construct validity via the known group procedure as well as and internal consistency of α = 0.88 have been reported [69].

The Beck Anxiety Inventory (BAI) [70] is a widely used standardized self-report questionnaire aiming to assess the severity of anxiety symptoms among adolescents and adults based on the DMS-IV anxiety criteria in adulthood. This questionnaire consists of 21 items rating the degree to which one is bothered by the symptom described. Each item is rated on a four-point Likert scale ranging from 0 (not at all) to 3 (severely, I can barely stand it). A total score ranging from 0 to 63 is calculated by summing the severity ratings for all 21 items. Excellent internal consistency (Cronbach’s alpha = 0.92) and a test–retest reliability coefficient of 0.75 have been reported [70,71,72] Criterion validity was tested utilizing the clinician-rated Hamilton Anxiety Rating Scale—Revised [73] (r = 0.51).

The Sensory Responsiveness Questionnaire—Intensity Scale (SRQ–IS) [74] is a standardized self-report questionnaire aiming to identify sensory modulation dysfunction amongst adolescents and adults. Comprising 58 items, the SRQ-IS assesses responses to sensations experienced in daily life, each representing a specific sensory modality (i.e., auditory, visual, gustatory, olfactory, vestibular, and somatosensory stimuli). The items are phrased in an Aversive/Hedonic manner, (e.g., *I am bothered by the way new clothes feel; I am bothered by background humming noises (air conditioner, refrigerator, computer fan*), and responders are required to rate the intensity of their responses. Using a 5-point Likert scale ranging from 1 (not at all) to 5 (very much), the SRQ-IS provides 2 scale scores: SRQ-Aversive (32 items), representing high sensory responsiveness, and SRQ-Hedonic (26 items), representing low sensory responsiveness, where higher scores indicate higher/lower sensory responsiveness, respectively. Internal consistency indicated a high Cronbach’s alpha (0.90, 0.98 for the Hedonic and Aversive scales, respectively), as well as criterion and construct validity [74].

A demographic and behavioral questionnaire was developed for this current study, inquiring about mothers’ age, marital status, education, socioeconomic status, and their children’s age and grade. In addition, mothers were requested to rate their child’s performance on homework preparation and social functioning using a 10-point scale ranging from 1 (significant difficulties) to 10 (no difficulties). Additionally, their child’s social, behavioral, and organizational difficulties were queried.

### 2.3. Procedure

An online survey was distributed through social media to mothers using the Qualtrics platform, in one questionnaire sequence which took approximately 15–20 min to complete.

### 2.4. Data Analysis

Using IBM SPSS Statistics version 24, descriptive statistics were used to describe the sample and variables. The Shapiro–Wilks test was used to test for normality. Group differences were tested for using independent-samples t-tests or chi-square tests for categorical variables. Within each group, correlations were assessed between all variables using Spearman correlations. Correlations were considered low (ranging from 0.25 to 0.49), moderate (0.5 to 0.75), or high (>0.76) (Portney, 2020). A significance level of *p* < 0.05 was set for all tests. The magnitude of group differences was assessed by calculating Cohen’s d-effect size as small (0.1), medium (0.3), or large (0.5) [75].

## 3. Results

Sixty-seven mothers of children with ADHD (N = 40; mean (SD) age 40.9 (5.6) years)—henceforth called the study group—and without ADHD (N = 27; mean (SD) age 42.5 (5.0) years)—henceforth called the control group—participated in this study. No statistically significant differences (*p* > 0.05) were found between groups in terms of age, education level, marital status, socioeconomic status, or their child’s age (see Table 1).

### 3.1. Group Differences in Study Measures

A statistically significant group difference was found in parental self-efficacy; the study group reported having a lower self-efficacy compared to the control group, indicating a nearly large effect size (Cohen’s d = 0.71). Further, the groups differed in the SRQ-Hedonic score; the study group reported statistically significantly higher scores, showing a moderate effect size. No statistically significant group differences were found in the BRIEF-A (on the GEC and both indexes BRI and MI), SRQ-Aversive scale, or the BAI (see Table 2).

### 3.2. Within-Group Correlations between Study Measures

Within the study group: The BRIEF-A indexes were moderately correlated with the BAI scores; higher scores (lower EF) were found to be significantly correlated with elevated anxiety. In addition, the BRIEF-A indexes and the SRQ-Aversive score were found to be moderately correlated. Thus, lower EFs (higher scores) were significantly correlated with higher sensory responsiveness (Table 3).

Within the control group: The BRIEF-A indexes were moderately correlated with the BAI scores; higher scores (lower EF) were found to be significantly correlated with elevated anxiety. The BRIEF-A indexes were negatively correlated with the SRQ-Hedonic scale; thus, lower scores (better EF) were found to be significantly correlated with higher SRQ-Hedonic scores (low sensory responsiveness). The BRIEF-A BRI index was negatively correlated with the Parental Self-Efficacy scores; thus, lower scores (better EF) were significantly correlated with a higher perception of parental self-efficacy (Table 3).

### 3.3. Group Differences in Children’s Performance

Statistically significant group differences were found in children’s homework preparation and social abilities, as well as in their behavioral, emotional, and organizational skills difficulty (Table 4). Mothers of children with ADHD reported that their children are less independent in homework preparation, have a lower social ability, and greater behavioral, emotional, and organizational difficulties compared to children in the control group (children without ADHD).

## 4. Discussion

This preliminary study aimed to explore EFs, self-efficacy, anxiety, and sensory modulation in mothers of children with ADHD, attempting to better understand factors affecting their co-regulation abilities [25,36,50,76,77]. This study found enhanced sensory under-responsiveness in mothers of children with ADHD. Congruent with previous reports [53,78], we also found reduced parental self-efficacy in mothers of children with ADHD, additionally supporting previous reports indicating reduced self-efficacy among adults with ADHD in general [79,80,81]. It may be suggested that enhanced sensory under-responsiveness supports attenuated self-efficacy in mothers of children with ADHD, since sensory under-responsiveness implies the affected individual is missing out more subtle environmental stimuli [55]. Overlooking or missing cues may cause in-situation maternal reactions which are lacking the full context and therefore not efficient and succinct, thus eliciting a self-perception of compromised mothering performance, which is possibly expressed through reduced self-efficacy. This study also found that children with ADHD show multiple challenges across social, behavioral, emotional, and organizational functions and are significantly less independent in homework preparation, supporting previous reports [49,82] and alluding to the familial toll that ADHD poses [19,20,24,25].

The abovementioned challenges may yield familial strains, elevated stress, and reduced parental competency, as previously reported [24,83]. As parents of children with ADHD are confronted with greater day-to-day challenges linked to this [83], these challenges may also potentially contribute to the reported attenuated self-efficacy in mothers. Furthermore, lower parental self-efficacy can be viewed as an obstacle affecting children’s functional and regulation capabilities [24]. Thus, a better understanding of factors linked to parental self-efficacy is crucial to solving this chicken–egg riddle.

This study did not find significant differences in executive functions, anxiety, or sensory-aversive responses between mothers of children with and without ADHD. Further, amongst mothers of both groups, executive functions and anxiety traits were linked, demonstrating that lower executive function abilities are related to elevated anxiety traits, indicating in both groups that deficiencies in abilities such as initiation, working memory, and planning and monitoring task regulation are linked to increased anxiety. It may be suggested that inefficient executive functions to master daily activities induces elevated stress in mothers’ day-to-day interactions with their children [84]. In turn, higher anxiety and stress may be linked to affected EFs, primarily metacognitive and working memory abilities [85]; interestingly, both contribute to parental self-efficacy [24]. Although we found a similar correlation pattern in both groups, it is worthy to note that ADHD in children impacts both the children’s and parents’ daily functioning and induces higher stress levels in mothers [24,25,84]. Indeed, EFs and anxiety traits were found to mutually impact co-regulation in children with ADHD [5,35,37,40,86].

Among mothers of children with ADHD, we found that higher sensory *over-responsiveness* is linked to reduced EF abilities. Interestingly, in the control group (mothers of children without ADHD), we found that higher sensory *under-responsiveness* is linked to higher EF abilities. This is the first study reporting these ties among mothers of children with ADHD, which may support previous reports demonstrating that adults with specific difficulties (e.g., learning disabilities or ADHD) and reduced executive function abilities also exhibited sensory modulation difficulties and compromised self-regulation capabilities [11,87,88]. Considering that sensory modulation is a key factor in the ability to regulate oneself [89,90,91,92], it is possible that genetic predispositions and environmental factors are at play [16]. Findings may lead to better understanding the role of sensory modulation in parental scaffolding of children, from both perspectives—modeling as well as co-regulation [16]. While previous reports indicated that sensory modulation and anxiety are related [56,93,94], sensory modulation and executive functions have also been reported to co-occur, affecting emotional regulation, self-control, and daily activities [87,88,95,96].

Appropriately managing ADHD symptoms utilizing EF strategies and self-regulation in children with ADHD emphasizes the need for parental intervention. The current clinical practice guidelines for the treatment of children ages 6–12 with ADHD include the use of medications along with parent training and/or behavioral intervention, preferably both [97]. Further, recent recommendations encourage parent involvement in interventions for children with ADHD [98,99]. Indeed, incorporating parents has been shown to increase intervention gains and carry-over from the clinic to the home. Understanding parental ADHD-related difficulties as a risk for more severe clinical presentation of ADHD symptoms in children as well as family conflicts and co-regulation challenges has paramount implications when considering intervention [100]. Moreover, ADHD-related difficulties in mothers may attenuate their self-efficacy, affecting co-regulation and scaffolding capabilities [101]. Interventions such as Parental Occupation Executive Training (POET [102]), in which parents participate in weekly one-on-one sessions, promote personal daily functioning goals for young children with ADHD. POET has been shown to be effective in early stages of research, demonstrating improvement in parents’ ability to self-regulate and suggesting an enhancement in their ability to support their child’s executive functions and daily functioning [102]. In addition, the Cog-Fun OT intervention suggests that parental self-efficacy highly benefits from parental involvement, beyond the improvement in the child’s daily functioning and executive functions [52]. While the parental perspective is fundamental for therapy success [98], we found no intervention aimed at improving parents’ own sensory modulation and emotional regulation as part of a parental intervention. Our findings provide a proof of concept with potential benefits for the daily functioning as well as the family relationships of children with ADHD in both research and practice.

### 4.1. Study Limitations and Future Research

This study had several limitations. As a preliminary study, we used a small sample size comprising mothers only. In addition, data on past or present treatments or counseling, despite being common, were not requested and thus might influence the findings. Increasing the sample size and widening the scope to include both parental views and previous interventions may yield a better understanding of family dynamics. Further, this study overlooked mothers’ own ADHD diagnosis as well as personality traits, psychiatric conditions, and major life events, which may have affected the findings. To deepen the understanding of EFs in mothers of children with ADHD, future studies ought to consider those confounding factors. In addition, in the current study, the nature and source of children’s ADHD were not investigated, which may limit the understanding of ADHD manifestation and its implications. Future studies should identify the source, type, severity of the ADHD diagnosis and comorbidities, and the level of difficulties caused by ADHD in boys vs. girls. This will allow a better understanding of the mothering of children with ADHD.

### 4.2. Conclusions

Mothers of children with ADHD demonstrate lower self-efficacy and lower sensory responsiveness. Further, in mothers of children with ADHD, less sufficient EF abilities were linked to higher sensory responsiveness, anxiety, and lower parental self-efficacy. Thus, self-efficacy, executive functions, high sensory responsiveness, and anxiety are uniquely interwoven in mothers of children with ADHD and may interfere with family relationships. These should be addressed to broaden research and clinical perspectives targeting co-regulation in families with children with ADHD.

## Figures and Tables

**Table 1 children-10-01286-t001:** Demographic characteristics of the two groups of mothers.

	Study Group (N = 40)	Control Group (N = 27)
Mothers’ age (years) Mean (SD) Min–Max	40.9 (5.6) 30–52	42.5 (5.0) 29–54
Child’s age (years) Mean (SD) Min–Max	9.0 (1.89) 5–12	9.0 (1.93) 5–12
	**N (%)**	**N (%)**
Level of Education		
High school	6 (15%)	2 (7.4%)
Undergraduate studies	14 (35%)	7 (25.9%)
Graduate studies	19 (47.5)	14 (51.9%)
PhD or a similar degree	1 (2.5%)	4 (14.8%)
Marital Status		
Married/In a relationship	37 (92.5%)	24 (88.9%)
Single parent	1 (2.5%)	1 (3.7%)
Separated	2 (5%)	2 (7.4%)
Socioeconomic Status (%) *		
Below average income	4 (10%)	6 (22.2%)
Average income	11 (27.5%)	4 (14.8%)
Above average income	25 (62.5%)	17 (63%)

* According to government statistical data.

**Table 2 children-10-01286-t002:** Group differences in mothers’ executive function, anxiety, parenting self-efficacy, and sensory modulation scores.

Measure (Score Range)	Study Group (N = 40)	Control Group (N = 27)	Differences between Groups		
	Mean (SD) Min–Max	Mean (SD) Min–Max	*t*	*p*	Cohen’s d
BRI (30–100)	51 (10.3) 31–72	49.8 (10.6) 33–72	0.4	0.7	0.115
MI (30–100)	68.6 (19.6) 42–109	67.2 (14.8) 43–109	0.4	0.7	0.081
GEC (60–179)	120 (28) 96–144	117 (23) 100–128	0.4	0.7	0.117
BAI (21–66)	31.7 (5.4) 21–60	29.4 (5.6) 21–39	1.1	0.3	0.418
Self-Efficacy (1–6)	4.2 (0.7) 2.9–5.9	4.7 (0.7) 2.0–5.5	−2.4	0.018	0.714
SRQ-Hedonic Scale (1–5)	2.2 (0.4) 1.5–3.3	2.0 (0.3) 1.4–2.7	2.3	0.025	0.566
SRQ-Aversive Scale (1–5)	2.3 (0.4) 1.4–3.2	2.4 (0.3) 1.8–3.0	−1.8	0.08	0.283

BRI—Behavioral Rating Index; MI—Metacognitive Index; GEC—Global Executive Function Composite score; BAI—Beck Anxiety Inventory; SRQ—Sensory Responsiveness Questionnaire; SD—standard deviation.

**Table 3 children-10-01286-t003:** Within-group correlations between study measures.

BRIEF-A Indexes	Study Group N = 40				Control Group N = 27			
	BAI	SRQ-Hedonic	SRQ-Aversive	Self-Efficacy	BAI	SRQ- Hedonic	SRQ- Aversive	Self-Efficacy
BRI	0.65 **	0.15	0.39 **	−0.31 *	0.45 *	−0.35	0.18	−0.33
MI	0.41 **	0.05	0.34 **	−0.20	0.52 *	−0.38 *	0.16	−0.12
GEC	0.51 **	0.10	0.37 **	−0.26	0.60 *	−0.44 *	0.24	−0.24

* *p* < 0.05, ***p* < 0.01. BRI—Behavioral Rating Index; MI—Metacognitive Index; GEC—Global Executive Function Composite score; BAI—Beck Anxiety Inventory; SRQ—Sensory Responsiveness Questionnaire.

**Table 4 children-10-01286-t004:** Group differences (mean (SD) range/%) in homework preparation, social abilities, and behavioral, emotional, and organizational difficulties, according to mothers.

Children’s Performance (Score Range)	Study Group Mean (SD) Range (N = 40)	Control Group Mean (SD) Range (N = 27)	*t*-Test (*p*)
Homework Prep Independence (1–10)	4.1 (2.7) 1–10	7 (2.4) 1–10	0.5 (0.001)
Social Abilities (1–10)	6.0 (2.7) 1–10	7.8 (2.0) 2–10	−3.1 (0.002)
	**N (%)**	**N (%)**	**χ^2^ (*p*)**
Behavioral difficulty (Yes/No)	23 (56.1%)	6 (20.7%)	8.7 (0.003)
Emotional difficulty (Yes/No)	37 (90.2%)	14 (48.3%)	15.1 (<0.001)
Organization difficulty (Yes/No)	37 (90.2%)	8 (27.6%)	29.0 (<0.001)

SD—standard deviation.

## Data Availability

Data is available upon request.

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
