# Peer review of "Exploring Co-Regulation-Related Factors in the Mothers of ADHD Children—Proof of Concept Study"

_children, 2023, doi:10.3390/children10081286_

Round 1
Reviewer 1 Report
Of course, the role of parents in the upbringing of children with ADHD is significant, the success of the correction of this neurological disorder will directly depend on the contribution of both parents. Therefore, the mental and emotional health of parents is very important. Thus, the study conducted by the authors is relevant and interesting. However, during the article review, some observations arose:
1) The authors should indicate the degree of involvement in the process of raising mothers of both children with ADHD and mothers of children without ADHD. This is important, since the degree of participation of mothers in raising a child can be different (the mother can be with the child all day - if the mother does not work; the mother can be with the child for several hours a day - if the mother works; or maybe the father or grandmother is raising the child ). In the latter situation, it would be more correct to test another family member, since the mother is minimally involved in the upbringing process. The same goes for mothers of children without ADHD.
2) The article does not indicate the gender of children in both groups, which is also important. The authors should clarify this issue as well.
3) Authors should also indicate the severity of ADHD in children in the control group and the presence of comorbidities in them, as this also affects the emotional state of the mother. It is possible that some of the children included in the studies had disabilities, this should also be pointed out to the authors.
4) The article does not disclose the causes of ADHD, if the causes are of a genetic nature, then it is necessary to indicate which of the relatives was diagnosed with this. In this regard, it is also necessary to indicate the state of health and anamnesis of the mothers who participated in the study, since it is possible that the identified violations in the executive function of mothers are nothing more than the result of the presence of ADHD in the mothers of the subjects.
5) The results of the survey of mothers obtained by the authors are convincing, but controversial, since the possible influence of other external factors, such as family relationships, problems at work, loss of loved ones and the social status of the family, was not assessed. If these factors were evaluated by the authors, then these data should be included in the manuscript.
Author Response
Reviewer 1
We thank the reviewer for his/her time and effort. Your comments much improved our paper.
Of course, the role of parents in the upbringing of children with ADHD is significant, the success of the correction of this neurological disorder will directly depend on the contribution of both parents. Therefore, the mental and emotional health of parents is very important. Thus, the study conducted by the authors is relevant and interesting. However, during the article review, some observations arose:
1) The authors should indicate the degree of involvement in the process of raising mothers of both children with ADHD and mothers of children without ADHD. This is important, since the degree of participation of mothers in raising a child can be different (the mother can be with the child all day - if the mother does not work; the mother can be with the child for several hours a day - if the mother works; or maybe the father or grandmother is raising the child ). In the latter situation, it would be more correct to test another family member, since the mother is minimally involved in the upbringing process. The same goes for mothers of children without ADHD.
Response: Thank you for this comment. Mothering a child with ADHD vs. without ADHD is now elaborated and added into the background's second paragraph (Lines 15-27).
As proof of concept indeed we approached only mothers and this is pointed out in study limitations Lines 288-290.
2) The article does not indicate the gender of children in both groups, which is also important. The authors should clarify this issue as well.
Response: This study, focusing on mothers, aimed at delineating mothers of children with ADHD characteristics not previously studied together. Yet we also referred to their child's performance in daily routines. Thank you for this comment - for a deeper understanding of mothers of children with ADHD characteristics and daily strains it is worthy investigating boys vs. girls impact on mothering, and it is now added to future studies recommendation (Lines 296-298).
3) Authors should also indicate the severity of ADHD in children in the control group and the presence of comorbidities in them, as this also affects the emotional state of the mother. It is possible that some of the children included in the studies had disabilities, this should also be pointed out to the authors.
Response: In the control group there were no mothers of a child with ADHD nor any other disorder or illness. We included only mothers of healthy children.
As for the study group since this was an anonymous internet survey, we could not diagnose and determine the ADHD severity. We added this to future studies recommendation (Line 296-298). Of note, we demonstrated that in most study measures groups (mothers of children with ADHD vs. without ADHD) did not differ.
4) The article does not disclose the causes of ADHD, if the causes are of a genetic nature, then it is necessary to indicate which of the relatives was diagnosed with this. In this regard, it is also necessary to indicate the state of health and anamnesis of the mothers who participated in the study, since it is possible that the identified violations in the executive function of mothers are nothing more than the result of the presence of ADHD in the mothers of the subjects.
Response: Thank you for this comment. Indeed, mothers were not asked about their child's ADHD cause. This is now added (Lines 294-299). We did refer to the high prevalence of parental ADHD in children with ADHD (Lines 21-23), however we did not investigate the ADHD prevalence in mothers (Line 290-292). Please note that we did not find group differences in EF.
5) The results of the survey of mothers obtained by the authors are convincing, but controversial, since the possible influence of other external factors, such as family relationships, problems at work, loss of loved ones and the social status of the family, was not assessed. If these factors were evaluated by the authors, then these data should be included in the manuscript.
Response: Thank you for this comment. We now added Socioeconomic Status (Table 1 Lines 137-138; 163-166). Table 1 demonstrates that groups did not differ in socioeconomic status, education level, marital status as well as mother and child ages. In addition we have the BAI scores (measuring anxiety) which did not differ between groups. We have now added the above (yours) to future studies recommendation under major life events (see lines 290-292)

Reviewer 2 Report
The current study is focused on the comparative exploration of executive functions, self-efficacy, anxiety, and sensory modulation in mothers of children diagnosed with ADHD and in those of children without this disorder, using a case-control design. The conclusions of this article may be of interest to clinicians and other mental health specialists involved in supporting families of patients with ADHD. Still, some points need to be addressed in order to increase the fluency and accuracy of the data presentation- please see below:
Abstract
To avoid unnecessary repetitions, ”comparison group” may be replaced occasionally with „control group”, because there is no active intervention in this study. In the sentence beginning with „Further...”, note that „within” does not require a capital letter.
Background
Line 7- by „function” do you mean „executive functioning”, „daily functioning”, „cognitive function”, or something else?
Line 9, 11, etc.- it is redundant to mention between round brackets the detailed reference of a paper if that reference was already numbered and mentioned between square brackets;
Line 14- a round bracket is misplaced;
Lines 17-18- it is difficult to derive this conclusion from the previous sentence; consider rephrasing;
Line 19- did you mean „regulating behavior in children with ADHD”? Because „regulating children with ADHD” sounds strange;
Lines 32-35- please rephrase for clarity;
Line 47- „typical children...” would better be expressed as „children without ADHD...” or „healthy children”;
Line 48- „an intervention” or „interventions”;
Lines 79-82- any exclusion criteria? What about other inclusion criteria that should have eliminated potential confounders on the administered scales, e.g., psychiatric disorders that could impact the EF in mothers, or anxiety disorders; if such variables were not assessed, then these should be considered as limitations of the current study;
Line 80- how was the presence/absence of ADHD validated? Any structured methods? Did clinicians confirm the diagnosis of ADHD? Was the absence of ADHD based solely on mothers' statements?
Line 93- maybe consider a comma instead of a full stop, or, otherwise, rephrase because the sentence is incomplete;
Line 95- please rephrase „dysfunctional cognitive functions”; also, the construction „each specified scale both indexes” is incorrect;
Line 143- „where” is improperly used here;
Results
-important variables were not controlled, e.g., the presence of ADHD in mothers (since during the introductory part it is explicitly mentioned the high familial aggregation of this disorder); the existence of anxiety or mood disorders in mothers (because a psychological instrument was used to determine the level of anxiety); personality traits (other than anxiety as a feature) and disorders (these might have an impact on the self-image and self-efficacy); cognitive dysfunctions (in order to alleviate a potential impact on the EF evaluation); other diagnoses in children, that might have impacted their homework performance, like learning disorders or disorders of intellectual development; the presence/absence of treatment for ADHD symptoms, etc.
Proofreading is required; moderate English language errors exist.
Author Response
We thank the reviewer for her/his time and effort. Your comments much improved our paper.
The current study is focused on the comparative exploration of executive functions, self-efficacy, anxiety, and sensory modulation in mothers of children diagnosed with ADHD and in those of children without this disorder, using a case-control design. The conclusions of this article may be of interest to clinicians and other mental health specialists involved in supporting families of patients with ADHD. Still, some points need to be addressed in order to increase the fluency and accuracy of the data presentation- please see below:
Abstract
To avoid unnecessary repetitions, ”comparison group” may be replaced occasionally with „control group”, because there is no active intervention in this study. In the sentence beginning with „Further...”, note that „within” does not require a capital letter.
Response: Thank you, this was corrected along the abstract and paper as well.
Background
Line 7- by „function” do you mean „executive functioning”, „daily functioning”, „cognitive function”, or something else?
Response: Thank you. We added daily (Line 7).
Line 9, 11, etc.- it is redundant to mention between round brackets the detailed reference of a paper if that reference was already numbered and mentioned between square brackets;
Response: Sorry. Thank you
Line 14- a round bracket is misplaced;
Response: Thanks you. This was deleted
Lines 17-18- it is difficult to derive this conclusion from the previous sentence; consider rephrasing;
Response: This was rephrased " Thus, the parent-child relationship is at risk becoming fundamentally impacted by ADHD" (Line 22-23).
Line 19- did you mean „regulating behavior in children with ADHD”? Because „regulating children with ADHD” sounds strange;
Response: Thank you this was corrected (Line 28).
Lines 32-35- please rephrase for clarity;
Response: This was rephrased to "Therefore, sufficient parental EF are paramount, required to facilitate the child’s ability to self-regulate his behavior which in turn may elicit a better parenting performance." (Lines 41-44).
Line 47- „typical children...” would better be expressed as „children without ADHD...” or „healthy children”;
Response: This was changed to children without ADHD (Lines 57-58)
Line 48- „an intervention” or „interventions”;
Response: Sorry. Corrected. Thank you (Line 58).
Lines 79-82- any exclusion criteria? What about other inclusion criteria that should have eliminated potential confounders on the administered scales, e.g., psychiatric disorders that could impact the EF in mothers, or anxiety disorders; if such variables were not assessed, then these should be considered as limitations of the current study;
Response: Thank you for this important comment. Using an online survey, inclusion criteria were minimal. However we added this in the limitations and future studies section (Line 290-294).
Line 80- how was the presence/absence of ADHD validated? Any structured methods? Did clinicians confirm the diagnosis of ADHD? Was the absence of ADHD based solely on mothers' statements? Response: ADHD/ Without ADHD was solely indicated by the mothers' statements. Participation in the study was anonymous, thus no validation was conducted. Now this is added under Methods section (Line 88). In addition this is now added to the limitation section (Lines 296-299).
Line 93- maybe consider a comma instead of a full stop, or, otherwise, rephrase because the sentence is incomplete
Response: Thank you! This was corrected (Line 105).
Line 95- please rephrase „dysfunctional cognitive functions”; also, the construction „each specified scale both indexes” is incorrect;
Response: Thank you, this was rephrased (Lines 107-108)
Line 143- „where” is improperly used here;
Response: Thank you. This was deleted (Line 155)
Results
-important variables were not controlled, e.g., the presence of ADHD in mothers (since during the introductory part it is explicitly mentioned the high familial aggregation of this disorder); the existence of anxiety or mood disorders in mothers (because a psychological instrument was used to determine the level of anxiety); personality traits (other than anxiety as a feature) and disorders (these might have an impact on the self-image and self-efficacy); cognitive dysfunctions (in order to alleviate a potential impact on the EF evaluation); other diagnoses in children, that might have impacted their homework performance, like learning disorders or disorders of intellectual development; the presence/absence of treatment for ADHD symptoms, etc.
Response: Thank you for this comment. The variables mentioned above are of utmost importance for a deep understanding of mother characteristics impacting child behavior, and indeed are now mentioned in the limitation section (Line 290-299). Since, this is proof of concept online study, it is limited in its ability to provide deep understanding. However we demonstrate that while in most measures groups did not differ (including EF), mothers of children with ADHD have a different profile of inter-characteristic interplay compared to mothers of children without ADHD. We believe that our findings do provide proof, and indicate that a larger in-depth investigation is warranted.
Comments on the Quality of English Language
Proofreading is required; moderate English language errors exist.
Response: Thank you, The manuscript was re-edited.

Reviewer 3 Report
Dear Editor,
I really appreciate the opportunity to review the manuscript entitled:
"Exploring co-regulation related factors in mothers of ADHD children - A proof of concept"
I commend the authors for describing this critical and timely issue. The paper is interesting and well-written; however, I would like to highlight some issues that merit revision:
My only question is about any psychotherapeutic treatment or through counselors on the test subjects. This type of intervention could likely influence the results obtained and is particularly common although it is not often reported unless explicitly requested. I would ask the authors to add a brief note to this effect or if the data is not available to add it to the limitations
Author Response
We really appreciate your time and effort. Thank you for reviewing this paper.
Comment: I really appreciate the opportunity to review the manuscript entitled:
"Exploring co-regulation related factors in mothers of ADHD children - A proof of concept"
I commend the authors for describing this critical and timely issue. The paper is interesting and well-written;
Response: We thank you for these kind words
Comment: however, I would like to highlight some issues that merit revision:
My only question is about any psychotherapeutic treatment or through counselors on the test subjects. This type of intervention could likely influence the results obtained and is particularly common although it is not often reported unless explicitly requested. I would ask the authors to add a brief note to this effect or if the data is not available to add it to the limitations
Response: Thank you for this comment, we definitely agree. As proof of concept online study we did not request this information. This is now added to the limitation section (Lines 289-292).

Round 2
Reviewer 2 Report
The manuscript improved. Thank you for taking into account my recommendations.